# Challenges of Applying Simplified LCA Tools in Sustainable Design Pedagogy

**Suphichaya Suppipat** [1], **Kulthida Teachavorasinskun** [2] **and Allen H. Hu** [1,*]

1 Institute of Environmental Engineering and Management, National Taipei University of Technology, Taipei 10608, Taiwan; t107609404@ntut.org.tw
2 Department of Industrial Design, Chulalongkorn University, Bangkok 10330, Thailand; Kulthida.T@chula.ac.th
* Correspondence: allenhu@mail.ntut.edu.tw; Tel.: +886-2-277-12-171

**Abstract:** The growing recognition of the Sustainable Development Goals (SDGs) has been integrated globally into product design and business activities. Life cycle assessment (LCA) is considered a useful tool for designers to apply in the early stages of product design to mitigate the environmental impact. The study aims to identify the challenges of applying simplified LCA tools to improve the eco-efficiency of products and achieve a higher level of sustainable innovation. The study was conducted in a sustainable design course at Chulalongkorn University, Bangkok, for four consecutive years. All challenges and opportunities by using ECO-it, Eco-indicators, and the Materials, Energy use, and Toxic emissions (MET) matrix to assess the environmental impact in each phase of 11 home appliances are presented and discussed. Results show the positive potential of applying the tools to achieve function innovation in design for sustainable innovation. The needs for guided instruction, the availability of the database, the complexity of a study product, and the overlooking of social dimensions are four major challenges in applying the tools in the early stages of product redesign. Further study in testing the tools and developing a database in collaboration with industries should be conducted to compare and validate the results.

**Keywords:** sustainability; Ecodesign; life cycle assessment; sustainable innovation; sustainable design pedagogy

## 1. Introduction

The awareness of social and environmental problems has been increasing over the past decade. The 17 Sustainable Development Goals (SDGs) by the United Nations called for the action of all countries in 2015. SDG number 12 is trying to achieve environmentally sound management and efficient use of natural resources, as well as trying to ensure sustainable consumption and production patterns [1]. Thus, businesses and corporates have to pay closer attention to the design and development of products that improve the social and environmental performances of the entire product life cycle. Life cycle assessment (LCA) can be a helpful tool in Ecodesign if integrated into the company structure [2] and can be used to improve the environmental performance and determine the sustainability baseline of products [3,4]. LCA and Ecodesign can be combined simultaneously to examine and validate product redesign and decision making [5,6]. However, the study of Lindahl [7] showed that only 18% of engineering designer respondents applied LCA as a method or tool in their design for the environment in companies. This finding might depend on the tool's degree of usability and degree of appropriateness. Simplification of product development projects is needed in the practicality of environmental LCA methods and software tools used in the industry. Simplified LCA (i.e., streamlined LCA) was introduced to evaluate the environmental factors of a product, process, or service life cycle efficiently and provide the same type of results as a detailed LCA [3]. However, many of the current tools are very complex and not easy to use by designers [8]. Designers prefer methods and

tools that need minimal or no training and do not require additional time or resources to interpret [9]. A level of expertise is needed to accomplish environmental assessment from the designers' point of view and use the methods and tools more efficiently [9]. As a result, the LCA concept and simplified LCA tools should be introduced and trained in sustainable design education.

Radical changes and transformations of socio-technical systems such as system innovations for sustainability [10] and sustainability-oriented innovation [11] are needed to achieve sustainability. These transitions cover product and process innovations; changes in user practices, markets, policy, and regulations; infrastructure, culture, and lifestyle; and an organization's philosophy and values [10,11]. These innovation activities aim to contribute and realize the social and environmental value in addition to achieving economic returns for sustainability [10,11]. This study aims to identify the challenges in applying simplified LCA tools in improving the eco-efficiency of products and the possibility of achieving a higher level of sustainable innovation through action research by sustainable design classroom observation. The result of this in-classroom experiment is a proposal for further research in refining LCA-based tools for designers and improving the collaboration between sustainable design educators, design practitioners, and LCA experts.

The following section summarizes the relevant literature on simplified LCA methods and tools for Ecodesign that are used in this study and defines the four levels of design innovation for sustainability. The third section presents the materials and methods applied in this classroom research. The fourth section discusses the results. The fifth section presents the discussion. The sixth section elaborates the concluding remarks.

## 2. Background

In this section, six types of LCA-based tools for designers and product developers are presented. Three simplified LCA tools used in a practical manual of Ecodesign are introduced, especially emphasizing the analytical purpose of the tools and types of data. The four levels of design innovation for sustainability are also described, together with examples of electrical and electronic product cases.

### 2.1. Types of LCA-Based Tools for Ecodesign

Recent review works show that several case studies of LCA-based tools applied by designers in the Americas and European countries are published widely in the academic and gray literature [2–4,6,9,12–16]. However, supporting evidence of LCA-based tool application in other manufacturing countries, except in Japan [17,18], is still rare. According to timing in the tool application to different stages of the product development process, Bauman and Tillman [19] categorized LCA-based Ecodesign tools into six types: matrices, dedicated software-based LCA, ordinary quantitative LCA, LCA-derived rules of thumb and proxies, combination tools, and LCA as a creative tool.

### 2.1.1. Matrices

The tool is developed for designers to define the environmental hotspots of a product systematically by covering the main life cycle stages (i.e., production and supply of materials and components, in-house production, distribution, utilization, and end-of-life system) and environmental impacts in a straightforward manner [19]. A short descriptive statement about the materials used, recyclability, and major environmental impacts is required to fill such a matrix with information. Moreover, the quantitative information on the mass of materials used, energy consumption, and toxic emissions needs to be completed by providing them in absolute numbers or a range of scales. Cooperation between designers and environmental staff is recommended because the results are based on the knowledge of the designer and required expertise [9,19]. The matrix can be used during early (i.e., planning), intermediate (i.e., conceptual design), and late stages (i.e., embodiment design) of product development [19]. Good representatives of matrices with a life cycle perspective comprise the Materials, Energy use, and Toxic emissions (MET) matrix [6,12,20,21], the

abridged LCA matrix [22], the environmental design strategy matrix (EDSM) [23], and the environmentally responsible product assessment matrix (ERPA) [24].

### 2.1.2. Dedicated Software-Based LCA

This type of special LCA software tool, so-called "quick and dirty" LCAs, is developed to solve a crucial problem in a time-intensive issue of applying a full-scale LCA in product development [19]. These software tools allow the prompt execution of an LCA by providing built-in material and process databases (e.g., cradle-to-gate data) and many methodologies for designers and product developers. Moreover, according to a default of the LCA method, the results are shown as a single score (i.e., weighted results). Without using software tools, the numbers that express the total environmental load of various materials and processes can also be found in the standard Eco-indicator report available on the Internet [25]. This method helps designers and product developers compare and identify environmental strengths and weaknesses and investigate issues that need further improvement in the designs. Dedicated LCA software tools can be used during early (i.e., planning), intermediate (i.e., conceptual design), and later stages (i.e., detail design) of product development [19,26]. Examples of such dedicated software tools are ECO-it by PRé Consultants B.V. [12,27], EcoScan by NTO Industrial Technology [12,28], Environmental Priority Strategies (EPS) 2015 by GaBi Solutions [28–30], and Idemat by TU Delft [12,28].

### 2.1.3. Ordinary Quantitative LCA

For better quality to support in-depth analysis of environmental tradeoffs, an ordinary quantitative LCA should be undertaken and requires environmental specialist assistance [19]. Furthermore, this type of quantitative LCA can be used not only for product evaluations but also for defining rules of thumb and proxies. The tool implementation by Danish companies showed that it can enhance a broad average in the environmental improvement of products from 30% to 50% [31]. The ordinary quantitative LCA can be used during early (i.e., planning) and later stages (i.e., detail design) of product development [19]. Good representatives of this quantitative LCA are the environmental design of industrial products (EDIP) method and tools [31,32].

### 2.1.4. LCA-Derived Rules of Thumb and Proxies

According to experience from the ordinary quantitative LCA studies, LCA-derived rules of thumb are simple design rules developed from consecutive case studies with the same unique feature in the major source of the environmental impact of products [19]. Proxies are simple metrics that assess a product concerning its essential environmental properties [19]. The metrics show the cumulative weight over the life cycle of various materials and energy used. LCA-derived rules of thumb and proxies can be used during the entire product development process (i.e., planning, conceptual design, embodiment design, and detail design) [19]. Examples of such rules of thumb and proxies are material input per unit of service (MIPS) [33], embodied energy in the building industry [34], and the Ten Golden Rules [6,35,36].

### 2.1.5. Combination Tools

The highlight of combining LCA with other aspects of the assessment is to facilitate the designers and product developers with tradeoffs between environmental and other properties of the product such as technical performance or cost [19]. Bauman and Tillman [19] further explained that the combination tools are elaborate and simple due to the nature of the tools that draw upon various methodologies, but those methods and concepts are also already recognized by designers. The combined tools can be applied during intermediate stages in product development (i.e., conceptual design) [19]. Examples of such combined LCA tools are the eco-functional matrix [23,37], the environmental descriptors [6,38], and the combined quality function deployment (QFD) with LCA such as Green QFD [6,39].

### 2.1.6. LCA as a Creative Tool

Most of the LCA-based tools in the previous categories are an analytical tool. A good example of an LCA-based creative tool is the reverse LCA (RLCA) suggested by Graedel [40,41]. RLCA creates the task of defining a functional unit to identify the needs that the product is intended to serve more comprehensively. The tool focuses on the needs of environmental characteristics and function (i.e., human needs) rather than the physical design of products, and in this way, it enhances creative system thinking and encourages the discovery of innovation opportunities [19,41]. The LCA studies of reference products can be utilized as inputs for brainstorming sessions during the early stages of product development. Bauman and Tillman [19] stated that the environmental knowledge gained from the reference products possibly influences the designers' mental frame of reference and can be applied intuitively in the design process.

### 2.2. Simplified LCA Tools Introduced in a Practical Manual of Ecodesign

Simplified LCA is not intended as a comprehensive quantitative determination, but rather as a method to identify "hotspots" in the environment and highlight important opportunities for environmental improvements [3]. In the early stages of product development, it is considered an effective, helpful tool for eco-designers [3,19]. Various case studies showed the successful application of the simplified LCA methods to identify substantial environmental aspects and improve the eco-efficiency of electrical and electronic products [12,14,36,42,43]. Bauman and Tillman [19] mentioned different examples of simplified LCA tools such as life cycle-influenced matrices, LCA-derived proxies and rules of thumb, and software-based LCA tools. Three simplified LCA tools are introduced in "A Practical Manual of Ecodesign" published by Ihobe [12]: MET matrix, Eco-indicators, and software tools for LCA.

### 2.2.1. MET Matrix

The MET matrix is a qualitative or semiqualitative approach that is used in each phase of the product life cycle to achieve a global view of the inputs and outputs [12]. This tool aims to identify the most substantial environmental concerns during a product's life cycle, which can be used to establish various improvement strategies; classifying the environmental issues into categories is important [3,21]. The necessary prioritization of environmental aspects is qualitative and focused on the knowledge of the environment and a golden rule, not on statistics or figures [12]. The criteria for assessment are the material cycle, energy use, and toxic emissions [3]. "M" stands for utilization of materials in each stage of the product life cycle and refers to all the inputs (i.e., consumption) required [12]. "E" stands for the utilization of energy throughout the product life cycle, where the major impact is mainly from production and/or transportation [12]. Lastly, "T" stands for toxic emissions and refers to all outputs such as emissions, effluent, or toxic waste produced in the process [12].

### 2.2.2. Eco-Indicators

The Eco-indicator is a simple quantitative tool for designers and product managers; it is more precise than the MET matrix and considered a quantitative approach because the prioritization of environmental aspects is based on numerical calculations [12]. Eco-indicators were developed by a multidisciplinary team formed by forefront industries, scientists, and the Dutch government to evaluate the environmental impact caused by industrial input activities, focusing on the damage impact on ecosystem quality, resources, and human health [12,25]. As a consequence, tables of numeric values expressing the environmental impact according to the quantity or volume of each material, process, or transport have been obtained. These values are expressed in units of their own, called millipoints (mPt), which are not equivalent to any other standard measurement unit [12]. When applying Eco-indicators to the product, the quantitative data of all input activities and the related Eco-indicator scores in mPt of the three stages, namely, production, use,

and disposal, need to be filled in on the product life cycle sheet. At the end, the sum of all quantitative values in each stage is determined to quantify the total impact of the entire product life cycle. The higher the Eco-indicator score, the greater the environmental load [25]. The numeric values expressing the environmental impact of those common materials and processes are collected in advance [25], but obtaining the Eco-indicator scores requires a laborious process [12].

### 2.2.3. ECO-It

ECO-it is a simple software-based LCA tool for product design teams that does not require advanced or special environmental knowledge for users to operate [12]. The evaluation is based on the Eco-indicator 95 [44] method providing the values for environmental guidance, not absolute values [12]. It calculates the environmental impact and shows which stages of the product life cycle contribute the most. This tool is considered a quantitative approach and a quick screening tool for improving the environmental performance of the product. The software comes with over 500 ReCiPe environmental impact and carbon footprint scores for commonly used materials, production, transportation, energy, and waste treatment processes [27]. Moreover, the users can edit and create their own databases with different scoring methods by using ECO-edit software [27]. This tool is considered the most complex tool compared with Eco-indicators and the MET matrix [12].

### 2.3. Levels of Design Innovation for Sustainability

Sustainability-driven innovation was defined by Arthur D. Little [45,46] as the creation of new market space, products, and services or processes fostered by social, environmental, and sustainability issues. Likewise, sustainable innovation is a process where sustainability considerations (i.e., environmental, social, and financial aspects) are integrated into company systems from idea generation throughout research and development and commercialization [45]. Furthermore, sustainable innovation covers the spectrum of innovation levels from incremental to radical. The four levels of innovation can be identified in the context of environmental improvement as follows: (1) product improvement, (2) product redesign, (3) function innovation, and (4) system innovation [10,47].

### 2.3.1. Level 1: Product Improvement (Incremental)

At this incremental level, the existing product has solely a small progressive improvement by replacing material, refining the shape and form, reducing the product weight, reducing numbers of materials or parts, and restructuring parts. The improvements may focus on mitigating single environmental impacts for the existing product [10]. Examples of such product improvements are recycling plastic materials from used home appliances to make new products, biodegradable semiconductors, no-glue–no-screw products, portable washers (i.e., mini washing machines), and energy-efficient refrigerators.

### 2.3.2. Level 2: Product Redesign (Green Limits)

At this redesign level, the existing product has a major change in design but a limited level of improvement in technical feasibility. The product concept remains almost unchanged, but the product is completely rebuilt from an environmental life cycle perspective [10]. Examples include new washing machines with superior overall environmental performance, eco kettles, solar power chargers, hydroelectric battery lamps, air purifiers that use indoor plants for filter air, and power hand crack radios (i.e., self-powered emergency AM/FM radios).

### 2.3.3. Level 3: Function Innovation (Product Alternatives)

At this functional level, the existing product may be replaced by a new product or service that satisfies the same functional needs. The innovation is not limited to current product concepts but is connected to how the functional purpose is accomplished [10]. Product–service systems can be considered at this level. Examples of function innovative

products are online platforms for home appliance sharing, laundromats (i.e., laundry services), washer–dryer machines, smartphones, copier and printer leasing and service, all-in-one printers, and teleconferencing.

### 2.3.4. Level 4: System Innovation (Radical)

At this radical level, the existing product is designed for a sustainable society by applying systems thinking. A new system replaces the entire socio-technical system with its artifacts, structure, economic models, socio-cultural principles, and institutional framework [10]. Examples of such system innovation include mixed-use communities, smart home systems, smart cities, and the Internet of things network.

## 3. Materials and Methods

Thailand, one of the manufacturing countries, is also aware of environmental issues and attempts to improve the quality of its products. Input sustainable thinking and environmental awareness through design education are crucial. Thus, the study was conducted during 2015–2018 in the sustainable design course at the Department of Industrial Design, Chulalongkorn University, Bangkok. This section describes the context of participation, data collection, and data analysis cultivated from the application of simplified LCA tools during in-classroom activities for four consecutive years.

### 3.1. Participants

All participants were industrial design sophomores and juniors who registered for the sustainable design course during the second semester in 2015 ($n = 13$), 2016 ($n = 18$), 2017 ($n = 10$), and 2018 ($n = 7$). The sustainable design course is a three-credit elective course, comprising a one-hour lecture, four-hour independent study, and four-hour in-classroom activity per week, with a duration of 16 weeks. The study was conducted in the first four weeks of the course. In the first week of the study, 10 environmental impact categories, namely, eutrophication, acidification, photochemical oxidation, terrestrial ecotoxicity, marine aquatic ecotoxicity, freshwater aquatic ecotoxicity, human toxicity, ozone layer depletion, global warming, and abiotic depletion, were introduced. The participants learned what the root causes and major sources of each impact category were. Moreover, background knowledge about the LCA concept, product life cycle (i.e., extraction phase, manufacturing phase, packaging and distribution phase, use phase, and disposal phase), LCA methodology, and the definition of a functional unit was brought up and discussed. In the second week of the study, three simplified LCA tools for designers, namely, the MET matrix, Eco-indicators, and ECO-it, were inaugurated. The advantages and disadvantages of the tools were discussed and considered. Then, the assignment about simplified LCA tool application was delivered at the end of the one-hour lecture. The participants were trained to use those three different tools during the four hours of the in-classroom activity, and one of the authors as an instructor provided several examples and acted as a facilitator during a tutorial session. In the third week of the study, the instructor provided an overview of the Lifecycle Design Strategies wheel [20,48] and showed various product and service design examples that applied these strategies to ensure the participants can understand the overall design concept of holistic approaches. The four levels of design innovation for sustainability [11,45,46], the definitions of eco-efficiency, and factor X [43,49,50] were introduced. In the fourth week of the study, the participants were requested to present their results from the assignment and their reflections on the tool application.

The students were assigned to execute an Eco-redesign project by following the seven steps for implementation in "A Practical Manual of Ecodesign" published by Ihobe [12] and focus on the stage of environmental aspects to analyze the main environmental hotspots of the product throughout its life cycle. The task was to perform reverse engineering, assess the impacts of a small household appliance, and propose a redesign concept for reducing those impacts. Afterwards, all participants were divided into groups of two to four (depending on the total number of students in the class each year) and asked to

draw a product's life cycle diagram and disassemble a selected home appliance to build raw material inventory data. All parts and components were separated into groups and weighed based on different types of materials and processes for a further calculation to find the environmental impact contribution from the entire product life cycle perspective. First, a functional unit and a process tree of the selected home appliance based on the data collection and product disassembly were defined. Then, the environmental impact contribution (e.g., single score indicators) in each phase of the product's life cycle was calculated either manually or by software. A choice of the simplified LCA tools was limited to ECO-it software by PRé Consultants B.V. [12,27], Eco-indicator scores for materials and processes available in the Eco-indicator 99 Manual for Designers [12,25], and the MET matrix [3,12,20,21] because these tools are available publicly online and can be easily accessed. In addition, these tools are considered dedicated software-based LCA and matrix types that are normally applied in the early and intermediate stages (i.e., planning and conceptual design) of product development [19]. Lastly, they were requested to present a redesign proposal of impact reduction concepts, a list of problems and obstacles they encountered during the implementation of the tools, and any other food for thought and benefits they experienced by applying these tools.

### 3.2. Data Collection

The results from the reverse engineering, the impact assessment of all small household appliances, and the proposal of new redesign concepts were collected in the format of a design project report. The students had to submit one report per group at the end of the fourth week. All redesign proposals of impact reduction concepts in each group were presented in short sentences and sketches. The tool users' interactions with the tools and comments were derived from a participant observation during the second and third weeks of the tutorial session and the in-classroom activities. Reflective comments about the choice of the selected tool and their opinions on the tool application were also collected from the design reports and an online anonymous survey form.

### 3.3. Data Analysis

The participants were asked to express their opinion on user experiences and further suggestions for each tool that they used in a reflective comment format presented in the design project report of each group. Individuals were also asked to express their points of view of the tool application anonymously in the online survey form after their presentation. All comments were analyzed into two aspects of the tool application, namely, challenges and opportunities, and a common theme out of both aspects for discussion was identified. All of the new design proposals of impact reduction concepts were analyzed by mapping onto the levels of design innovation for sustainability proposed by Brezet [10,47]. By doing this, the authors can evaluate whether the simplified LCA tools can help the designers achieve system innovation in product design.

## 4. Results

The results are divided into two parts: the application of simplified LCA tools, and the challenges and opportunities identified from simplified LCA tools applied by industrial design students.

### 4.1. Application of Simplified LCA Tools

In 2015, 13 student participants were divided into two groups of four and one group of five. Three different kitchen appliances were chosen for the project, as shown in Table 1. In 2016, 18 student participants were divided into two groups of three and three groups of four. Four different home appliances were selected, as shown in Table 2. In 2017, 10 student participants were divided into five groups of two. Various types of household appliances including headphones, an iron, a sandwich maker, speakers, and a water heater were selected, as shown in Table 3. In 2018, seven student participants were divided into two

groups of two and one group of three. Three different small appliances were chosen for the design experiment, as shown in Table 4. After the students applied at least one of three different simplified LCA tools along with collecting their product inventory data of the chosen home appliance, the results from 16 groups of students with 11 different types of electrical and electronic products are shown in Tables 1–4. The 11 selected home appliances were (1) an electric pot, (2) a juicer, (3) a toaster, (4) a blender, (5) a sandwich maker, (6) a hairdryer, (7) an iron, (8) speakers, (9) headphones, (10) a CD player, and (11) a water heater. The most popular choice of simplified LCA tool application was Eco-indicators followed by ECO-it software and the MET matrix. The students defined a functional unit based on their user experience of the product including the frequency of use and the product lifetime. The highest environmental hotspot of electrical and electronic equipment came from the use phase due to high energy consumption. However, several of them that were not frequently used or used in a short period of time showed the highest impact in the production phase due to a considerable number of components and parts, such as the toaster, blender, speakers, and CD player.

According to environmental hotspot analysis and impact reduction concepts, the redesign proposals showed several relevant points in product design and the cause of the highest impact in the use phase (shown in italics in Tables 1–4). Examples include reducing the thickness of the material to allow the water to be heated quicker, changing the heating method, reducing the product size to match a functional unit, applying renewable energy, reducing the energy supply and time consumption during use, reducing the heat loss, finding alternative sources of energy, integrating the function with other products, redesigning the shape and form to enhance product performance, sharing the energy or electricity used with other products, and optimizing the energy used or the temperature to match a functional unit. However, considering the design proposal with the levels of design innovation for sustainability showed that the proposed concepts were still mostly at the levels of product improvement, product redesign, and function innovation. Only one group of students working on the water heater could propose a new design concept at the level of system innovation by introducing an idea of energy sharing for a home appliance cluster (see Figure 1).

**Table 1.** Application of simplified LCA tools in a product design process by industrial design students in year 2015 (*n* = 13).

| Home Appliances | Simplified LCA Tools | | | Functional Unit | Environmental Hotspot Based on Product Life Cycle | | | Hotspot Analysis | Design Proposal |
|---|---|---|---|---|---|---|---|---|---|
| | ECO-it | Eco-Indicators | MET Matrix | | Production | Use | Disposal | | |
| **Electric Pot** | | ✓ | | Boiling 2 L of water for 2 h, once a week for 4 years | | ✓ | | Needs a large amount of energy/electricity to heat the water | - Reduce the number of parts<br>- *Reduce the thickness of the material, and allow the water to be heated quicker*<br>- Change ABS plastic to other types of plastic because the Eco-indicator score of ABS is high<br>- Change to 100% recycled aluminum<br>- *Change the heating method*<br>- Redesign the form with less material, and contain the same volume of water |
| Juicer | | ✓ | | Squeezing oranges for 20 min, twice a week for 10 years | | ✓ | | Needs a large amount of energy/electricity to generate the motor | - *Reduce the product size to match a functional unit*<br>- Reduce the thickness of housing<br>- Change to other types of plastic because the Eco-indicator scores of ABS and PVC are high<br>- Reduce the number of parts or rearrange parts |
| Toaster | ✓ | | | Toasting bread for 5 min, once a week for 13 years | ✓ | | | Eco-indicator scores of aluminum, ABS, and steel for housing and inside parts are high | - Change the heating method<br>- Reduce the product size, the number of parts, and materials<br>- Rearrange the inside parts<br>- Use other types of plastic and 100% recycled aluminum instead |

**Table 2.** Application of simplified LCA tools in a product design process by industrial design students in year 2016 (*n* = 18).

| Home Appliances | Simplified LCA Tools | | | Functional Unit | Environmental Hotspot Based on Product Life Cycle | | | Hotspot Analysis | Design Proposal |
|---|---|---|---|---|---|---|---|---|---|
| | ECO-it | Eco-Indicators | MET Matrix | | Production | Use | Disposal | | |
| **Blender** | | ✓ | | Blending fruits for 3 min (making 300 mL of smoothie), twice a week for 5 years | ✓ | | | Eco-indicator scores of copper and steel for motor and inside parts are high | Change the blending method, take the motor away, or reduce the size<br>Reduce the product size<br><br>- Reduce the number, and rearrange the inside parts |
| Electric Pot | | ✓ | | Boiling 2 L of water for 2 h, once a week for 5 years | | ✓ | | Needs a large amount of energy/electricity to heat the water | - Redesign the structural joints without nut and bolt<br>- Use 100% recycled aluminum and biodegradable plastics instead<br>- *Reduce the product size to match a functional unit*<br>- *Change the heating method*<br>- *Apply renewable energy* |
| Hairdryer | | ✓ | | Drying damp hair for 15 min, 3 times a week for 5 years | | ✓ | | Needs a large amount of energy/electricity to heat and generate the motor | - *Change the hair drying method*<br>- *Reduce the energy supply and time consumption during use*<br>- *Reduce the heat loss* |
| Iron | | ✓ | | Ironing clothes for an hour, twice a week for 6 years | | ✓ | | Needs a large amount of energy/electricity to heat the soleplate | - *Change the ironing method*<br>- *Use other sources of heat, and reduce the energy consumption* |
| Speakers | | ✓ | | Listening to music for an hour per day, 5 days a week for 5 years | ✓ | | | Eco-indicator scores of copper and ABS are high | - Change the way of sound amplification<br>- Reduce the number of materials<br>- Avoid using copper, and change ABS to other types of plastic |

**Table 3.** Application of simplified LCA tools in a product design process by industrial design students in year 2017 (*n* = 10).

| Home Appliances | Simplified LCA Tools | | | Functional Unit | Environmental Hotspot Based on Product Life Cycle | | | Hotspot Analysis | Design Proposal |
|---|---|---|---|---|---|---|---|---|---|
| | ECO-it | Eco-Indicators | MET Matrix | | Production | Use | Disposal | | |
| **Headphones**  | ✓ | | | Listening to music for 3 h per day, every day for 3 years | | ✓ | | High energy/electricity consumption during use for the entire product lifetime | - Change to recycled ABS plastic<br>- Remove the metal used for a logo plate, and use the plastic engraving technique instead<br>- Reduce the nylon fabric used<br>- *Design to reduce electricity consumption, and shorten the time period during use* |
| Iron  | | ✓ | | Ironing clothes for an hour, 3 times a week for 5 years | | ✓ | | High energy/electricity consumption during use to heat the soleplate | - Use wrinkle-free fabrics<br>- *Shorten the use time* by spraying fabric softener mixed with water<br>- *Change the method of making the fabric smooth without using electricity* |
| Sandwich Maker  | | ✓ | | Making sandwich for 5 min per time, 5 times a week for 8 years | | ✓ | | Needs a large amount of energy/electricity to operate | - Use recycled materials<br>- Reduce the number of materials used<br>- *Integrate the function of the sandwich maker with other kitchen appliances (if possible)* |
| Speakers  | ✓ | | | Listening to music for 3 h per day, every day for 6 years | | ✓ | | High energy/electricity consumption during use for the entire product lifetime, high score of Eco-indicators for ABS | - Change ABS to other types of plastic<br>- Design without screws and easy to assemble<br>- Reduce the number of materials used<br>- *Redesign the shape and form to enhance sound amplification*<br>- Use recycled materials |
| Water Heater  | | ✓ | | Boiling 40 L of water for showering about 10 min per time, twice a day for 20 years | | ✓ | | High energy/electricity consumption during use to boil water, long product lifetime | - Use recycled materials<br>- *Share electricity/energy used for boiling water with other home appliances*<br>- *Use renewable energy such as solar power*<br>- *Control the water temperature, not too high than necessary* |

**Table 4.** Application of simplified LCA tools in a product design process by industrial design students in year 2018 (*n* = 7).

| Home Appliances | Simplified LCA Tools | | | Functional Unit | Environmental Hotspot Based on Product Life Cycle | | | Hotspot Analysis | Design Proposal |
|---|---|---|---|---|---|---|---|---|---|
| | ECO-it | Eco-Indicators | MET Matrix | | Production | Use | Disposal | | |
| **CD Player**  | ✓ | | ✓ | Listening to music for 3 h per day, every day for 2 years | ✓ | | | High impact from printed circuit board (PCB) and the number of alkaline batteries used | - Use recycled and recyclable materials<br>- Reduce the number of alkaline batteries used by using other sources of energy or using renewable energy such as solar power |
| Hairdryer  | | ✓ | ✓ | Drying damp hair for 30 min, every other day for 1 year | | ✓ | | Needs a large amount of energy/electricity to operate | - Use recycled materials<br>- *Reduce the use time by applying a dry hair towel*<br>- *Integrate the function with other home appliances to generate hot air*<br>- *Use renewable energy such as solar power* |
| Iron  | | ✓ | ✓ | Ironing clothes for an hour, once a week for 1 year | | ✓ | | High energy/electricity consumption during use to heat the soleplate | - Use recycled materials<br>- *Redesign the way to smooth the fabric without using electricity*<br>- Use wrinkle-free fabrics |

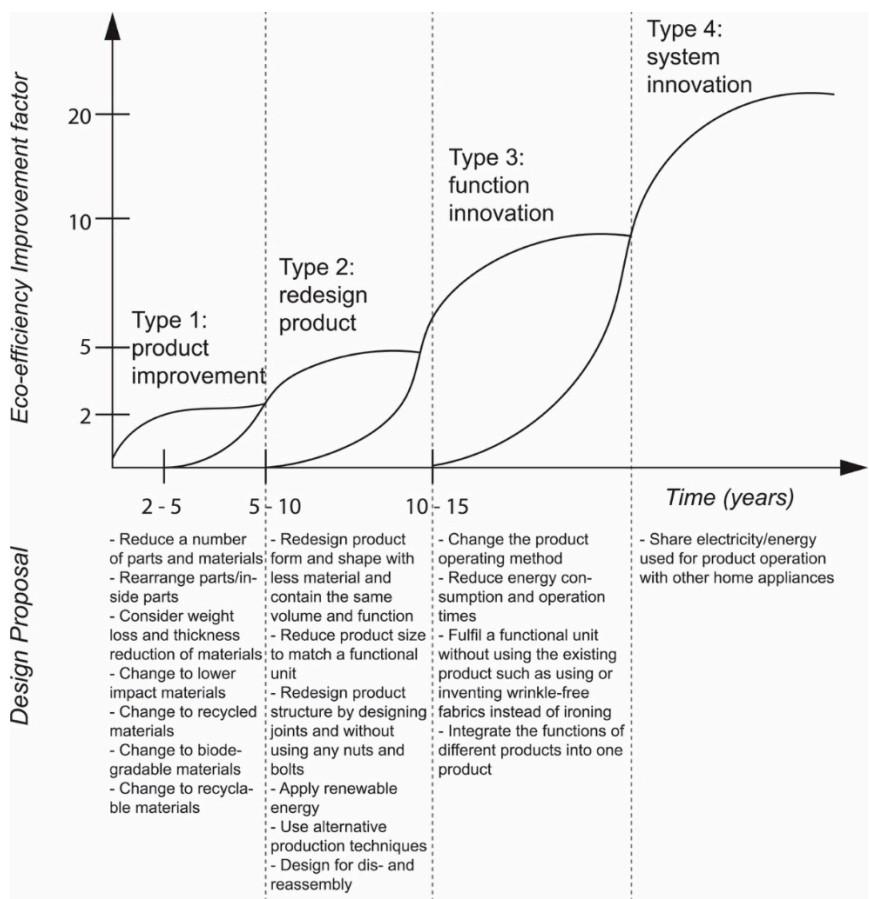

**Figure 1.** Mapping new design proposals of impact reduction concepts onto the levels of design innovation for sustainability (modified from Brezet [10,47]).

### 4.2. Challenges and Opportunities Identified from Simplified LCA Tools Applied by Industrial Design Students

According to the students' reflections, several challenges and opportunities were observed in the application of three different simplified LCA tools during product design and development, as shown in Table 5. First, time consumption in data input when applying ECO-it software and in searching for the appropriate database in the software was an issue, while the tool users pointed out that the software facilitated them in the calculation, iteration, and comparison of the results. Second, the tool users complained about the availability of Eco-indicator scores for materials and processes that were still missing, namely, the production of composite materials (e.g., mica plate, polytetrafluoroethylene), brass, porcelain, nylon, silicone, and road transport by motorbike. Furthermore, math skills were highly required. The miscalculation in unit conversion was a problem and affected the result. However, applying the table of Eco-indicator scores helped them discover environmental hotspots based on the product life cycle and present the factors that markedly contributed the most to the environment. Moreover, the Eco-indicator scores in mPt reviewed a wide range of material, production, and transportation information in the quantitative aspect that can be utilized to compare the impacts between the product's life cycle phases, materials, and processes more clearly and precisely. Third, the MET matrix, which was selected by only three groups of students, underlined the need for descriptive instructions of how to write the environmental evaluation into each cell of the matrix. The tool users were not sure what to write and how to describe each stage adequately and most of the time left the cells empty because some basic knowledge of materials science was required. However, due to the matrix structure, the tool users demonstrated that the matrix helped them look into the life cycle perspective and holistic

view of the product effectively and ensured the accuracy of their common sense about environmental problems. Overall, the common challenges in applying these simplified LCA tools for designers were how to derive and input the data such as the mass, weight, transportation distance, and product performance (i.e., quantitative and qualitative data) prior to conducting an environmental assessment and how to ensure that those data are adequate for evaluation. The results also revealed opportunities in the application of those tools such that designers or tool users can understand the big picture of the product's life cycle and think thoroughly to reduce the environmental impacts as well as compare different materials and processes for the final decision making.

**Table 5.** Reflections on user experience in the challenges and opportunities of simplified LCA tool application.

| Simplified LCA Tools | | Challenges | Opportunities |
|---|---|---|---|
| Quantitative approach | ECO-it | - The availability of databases is not very extensive for the moment.<br>- Entering the data into the software takes some time.<br>- Several parts are too small or too light to weigh and cannot be accountable for the assessment. | - Facilitate calculation and iteration.<br>- Possibly adapt and include this tool in the product design of a company.<br>- Enable comparing simple alternatives with the same type of product.<br>- Easier to use than expected. |
| | Eco-indicators | - Several parts are too small to weigh and cannot be accountable for the calculation. If several toxic substances are hidden, this may not be considered.<br>- Several parts are too small or too light to weigh.<br>- No Eco-indicator scores for composite materials such as the mica plate or chemical coating such as polytetrafluoroethylene (Teflon®).<br>- No Eco-indicator scores for brass, porcelain, nylon, and silicone.<br>- Requires math skills.<br>- Unit conversion problems as a result of miscalculation.<br>- Lack of a local database, for example, road transport by motorbike. | - Discover the environmental hotspots based on product life cycle.<br>- Present a wide range of information.<br>- Quantitative indicators can be used to compare the impacts between the product's life cycle phases more precisely and clearly.<br>- Show the factors that substantially impact or contribute the most to the environment considerably.<br>- Learn and compare the environmental impact of materials from the indicator scores. |
| Qualitative approach | MET matrix | - Requires basic knowledge of materials science.<br>- Not sure what to write and how to describe each stage properly. | - See more clearly from a life cycle perspective and a holistic view of the product.<br>- Can be used to recheck and ensure the accuracy of our common sense about environmental problems.<br>- Easy to apply. |

Regarding the participant observation in the classroom, several groups of students experienced difficulty during reverse engineering (i.e., disassembly session); for example, several of the electronic components cannot be disassembled by hand (e.g., printed circuit board [PCB], motor, plug power cord, light-emitting diode [LED], fuse, thermostat, and adapter), and there are several unknown materials, especially plastics and composite materials. These challenges raised an issue on how crucial the importance of design for disassembly is for further material recovery and waste management aspects that designers need to consider. Furthermore, after an hour of three different simplified LCA tool trainings, the participants showed a good ability to use the tools during the in-classroom activity. Several positive comments were raised during the tool implementation such as "the software tool is more user friendly than expected," "applying the tools helps to generate new ideas," and "the Eco-indicator scores broaden some environmental knowledge about materials and processes." However, several complaints, namely, how to define the

right functional unit, intensive time consumption in data collection, and complicated steps in data input to the software, were noted during tool application.

## 5. Discussion

In this section, the crucial points of this study are discussed considering four aspects of applying simplified LCA tools in sustainable design pedagogy: the opportunities, the challenges, the further suggestions, and the limitations.

### 5.1. Opportunities

The highlight of the practical application of simplified LCA tools from the design students' reflections is to focus on and emphasize the environmental hotspots based on the life cycle perspective that designers might ignore such as the use phase and the end of life of the products. The design students can identify the main negative contribution in each phase of the product's life cycle and compare the impact between different materials and processes by looking through the Eco-indicator scores to propose impact reduction concepts of a new product design proposal accordingly. The study of Bright and Boks [9] showed designers' supporting opinions on the utility of LCA in Ecodesign, namely, identifying remarkable environmental aspects, evaluating environmental trade-offs, and being useful in creativity and early design phases. The result of this study is in agreement with that of the study of Bright and Boks [9] that the application of simplified LCA tools can help design students propose new design concepts in an incremental change. They are mostly based on the level of product improvement, product redesign, and function innovation, which are not yet represented in a radical change in the level of system innovation. The tools assist in delicately redesigning product structure and performance, especially in energy consumption during the use phase. This positive feedback represents the importance of how simplified LCA tools can enhance design performance in the environmental aspect of sustainability. The study of Piekarski et al. also supported that the integration of the LCA software tool within the design course is inspiring and useful for students' future careers [51]. Previous studies of Bright and Boks [9], Rio et al. [26], and Piekarski et al. [51] addressed a crucial requirement of collaboration between designers and LCA experts to foster the systematically integrated environmental aspects into the design process. The need of adhering to a cross-industry innovation process can be emphasized to improve the performance of environmental innovation, such as new business models, new ventures and spin-offs, and new markets of technology application [52]. New design proposals at the level of system innovation can lead to system, institutional, and societal changes.

### 5.2. Challenges

The results of the literature review showed several obstacles in the application of LCA-based tools such as time limitation, defining the functional unit, and limited utilization in product design [2,7,9,35,53]. These issues were also mentioned in the students' reflections and emerged from the in-classroom activity observation. Moreover, the students shared several difficulties in disassembling parts and components of the selected home appliances (i.e., PCB, motor, plug power cord, LED, fuse, thermostat, and adapter) and finding the right Eco-indicator scores for unknown materials and processes. Furthermore, taking PCBs and LEDs apart was the most troublesome activity because they could not be separated manually by type of material. Special separation techniques such as shredding and crushing are required in practice at the end of home appliances' life. Therefore, unknown plastics and other materials were assumed for the assessment, which might cause an error in the results of the environmental impact because several embedded toxic substances cannot be considered. For example, an Eco-indicator score of brass was not in the Eco-indicator 99 Manual for Designers [25] but was categorized in a group of other nonferrous metals (e.g., zinc, brass, and chromium) in the Eco-indicator 95 Manual for Designers, with the score ranging from 50 to 200 mPt [44]. The wide range of scores and unknown embedded substances can affect the assessment result and seems an unavoidable problem. In this

case, the availability of the database related to materials, manufacturing processes, and transportation modes is a noteworthy issue. Some common materials, labor-intensive processes, and short-distance transportation options cannot be found yet in the simplified LCA database. The social aspect of sustainability also cannot be tackled sufficiently by applying the three simplified LCA tools. Nevertheless, the matrix types tend to have a high potential to integrate a social issue into consideration during the design process. The tools that can enhance systems thinking and a holistic perspective of sustainability, especially the social aspect, are crucial to achieve the level of system innovation.

*5.3. Further Suggestions*

The obstacles in the parts and components disassembly of the home appliances clearly review the crucial requirement in design for standardization, design for disassembly, and modularity in design for electrical and electronic products. Moreover, the Eco-indicator scores of the electronic components such as the LED, plug power cord, and PCB should be presented in an amount in mPt per unit or per standard size of the components instead of per mass of the components in kilograms. Furthermore, the information about new materials, processes, and transportation modes should be included and possibly updated in the simplified LCA database by the tool users. On this basis, designers can easily access the database and select appropriate data for the impact assessment. Given these obstacles, products with various numbers of components, such as electrical and electronic products, might not be a recommended choice for a design student or a beginner to learn how to apply LCA-based tools in practicing the Ecodesign process. Frequency and duration of practice are required to improve the learning curve of the simplified LCA tools. Moreover, some other simple products should be reconsidered for the first-time implementation. For example, painting brushes were applied in the LCA-based Ecodesign teaching practice of Piekarski et al. [51].

The study of Bright and Boks [9] underlined the design team needs; these requirements are a designer-friendly LCA-based tool, a tailor-made tool to meet a design team's specific needs across varying industries, and a tool to help interpret the LCA results as important guidelines in the designer's language. Various simplified LCA tools have been developed to match the design team's needs [3,6,19]. For software-based LCA tools, a framework of resolving interfaces between usual design and Ecodesign software was presented [26], and a university–industry collaboration of introducing a real case in LCA with Ecodesign considerations was proposed [51]. The three simplified LCA tools used in this study perform with a good designer-friendly interface and a potential to be applied with various types of products. However, they do not yet cover the need for facilitation in interpreting LCA results and providing any designer's language guidelines. As a result, a design guideline at the different levels of sustainable innovation should be embedded and/or linked to the simplified LCA tools for designers to apply in sustainable design pedagogy and professional practice. By doing this, the LCA result can practically facilitate and fully support industrial designers in creating a design proposal to achieve a high level of system innovation.

Over the past two decades, the study of Adams et al. [11] showed that the foundations of sustainable business practice began to be established, as reflected in many prominent environmentally and socially concerned platforms and initiatives for a business being set up and created. The research results of Skordoulis et al. [52] revealed a moderate level of environmental innovation implemented in firms. The most implemented practices were the ISO 14001 management systems and the toxic substances usage reduction. However, environmental product innovation was found to be the least implemented practice [52]. To enhance a positive perspective for competitive advantage development such as improving the reputation and customers' views of firms, more attention in environmental education should be paid to integrating innovation into their strategy and performance improvements [52]. Bright and Boks [9] also supported that LCA can be of good use in Ecodesign if it is embedded into the company structure. Firm competitive advantage may be possibly

enhanced by pursuing open innovation (i.e., using knowledge from internal and external environments of the firms) through co-creation, mitigation of climate change and environmental impact, and life cycle and energy efficiency by generating the university–industry relationship to improve simplified LCA and Ecodesign in product and process innovation. Different firm sizes tend to affect types of LCA tool implementation. The simplified LCA tools seem to be an effective starting point for small- and medium-sized firms.

### 5.4. Limitations

Only three simplified LCA tools were implemented in this study. The result from this implementation more or less cannot be representative of all simplified LCA tool types. Furthermore, the Eco-redesign project was executed and applied only to electrical and electronic products. The other products in some other industries, such as textile, food, and automotive industries, should be considered a case study in the future. The other limitation is that the total number of participants ($n = 48$) for four consecutive years and the variety of home appliances selected ($n = 11$) in this study were quite low. Further research design, data collection, and analysis of the simplified LCA tool testing with Ecodesign among newly recruited design students must continue to be implemented to compare and validate the results. Finally, this implementation was conducted in an academic setting only. The tool implementation with a real case and an academia–industry connection should be considered, as in the study of Piekarski et al. [51], in order to reveal the results.

### 6. Conclusions

When introducing three different simplified LCA tools, namely, ECO-it software, Eco-indicators, and the MET matrix, to industrial design students for application in the Eco-redesign project, 12 out of 16 groups of the students selected Eco-indicators to assess the life cycle impact of the chosen products and complete the assigned tasks. The 11 selected home appliances were evaluated for their environmental hotspots. The results show that seven of them (i.e., electric pot, juicer, sandwich maker, hairdryer, iron, headphones, and water heater) have the highest impact in the use phase, three of them (i.e., toaster, blender, and CD player) have a substantial impact in the production phase, and one of them (i.e., speakers) has the highest impact in either the use or production phase depending on the duration of use and product lifetime defined in the functional unit. New redesign proposals represent a positive potential of applying the simplified LCA tools to achieve the level of function innovation in sustainable innovation. However, the effectiveness to reach a higher level at system innovation has not been well achieved yet. Four major challenges remain in improving the eco-efficiency of the products in the early stages of development by applying the simplified LCA tools in sustainable design pedagogy. First, the needs for tool training and providing descriptive instruction are necessary for the first-time environmental evaluation of the products. Second, an issue in data preparation and input is the most time-consuming process. The availability of the database related to new types of materials, labor-intensive manufacturing processes, and short-distance transportation modes is also required. Third, the complexity of a study product can affect the precision of data selection and LCA results. Finally, the social aspect of sustainability might be overlooked in design concept generation due to some concerns of calculation skills and basic knowledge of material selection required during the impact assessment procedure. By applying the tools in the context of sustainable design pedagogy, the design students can gain various benefits in many ways, such as facilitating them to consider environmental problems delicately in all phases of the product life cycle and realizing the importance of design for disassembly, design for standardization, and modular design in electrical and electronic products, as well as improving their environmental and social critical and analytical skills to achieve a radical change in the level of system innovation. Further study on tool testing with newly recruited industrial design students should continue to be implemented to compare and validate the results. Moreover, the other types of products, a real-world case application, and an academia–industry partnership

should be considered for future study of the simplified LCA tool implementation. Finally, a collaboration between industrial designers, scholars, and LCA specialists' communities should be symbiosed to create more effective, user-friendly ways of LCA implementation.

**Author Contributions:** Conceptualization, S.S. and K.T.; methodology, S.S.; formal analysis, S.S.; investigation, S.S.; resources, K.T.; data curation, S.S.; writing—original draft preparation, S.S.; writing—review and editing, S.S., K.T., and A.H.H.; visualization, S.S.; supervision, A.H.H.; project administration, K.T. All authors have read and agreed to the published version of the manuscript.

**Funding:** This research received no external funding.

**Institutional Review Board Statement:** Not applicable.

**Informed Consent Statement:** Informed consent was obtained from all subjects involved in the study.

**Data Availability Statement:** The data presented in this study are available on request from the corresponding author. The data are not publicly available due to privacy restrictions.

**Acknowledgments:** The authors express their gratitude to the Department of Industrial Design, Chulalongkorn University for student participation and facility support, and the Institute of Environmental Engineering and Management, National Taipei University of Technology for financial support. The authors also thank KGSupport for their editing services.

**Conflicts of Interest:** The authors declare no conflict of interest.

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
