# Peer review of "Challenges of Applying Simplified LCA Tools in Sustainable Design Pedagogy"

_sustainability, doi:10.3390/su13042406_

Round 1

Reviewer 1 Report

Dear authors, thanks a lot for your manuscript! Although I consider the topic as relevant, I do not see yet enough academic insights in your paper to justify its publication. Following major drawbacks:

1) You did integrate too little review of existing literature on the topic of simplified LCA, but also its empirical investigation. The most interesting aspect is not only the easy integration of simplified tools, but the reliability and robustness. That is not discussed appropriately and was not part of your research design (could only be done by literature). Section 3.2 might point into that direction, but is way too short.

2) I do not understand why you add the types of innovation in Section 2.2 and Fig. 1 - you do not use that differentation in your discussion and conclusion section. You later introduce a concept of The Good, The Difficult, and the Suggestions - all by a sudden in Section 5. Where does that come from?

3) You tried to compare the three applied tools, but I do not see any relevant fact-based comparison: neither in your table nor text that allows readers/potential users to choose either ECO-it, MET matrix or Eco-indicators based on your result. You just state "the well-received choice was Eco-indicators due to the usability and the appropriateness of the tool within a limit of time and a certain task". I don't think these are scientific criteria for a appropriate choice.

4) You conclude that tools should integrate "local and cultural background information of the Asia manufacturing" (line 435): yes, that would be good, but it violates your advantage of simplified tools to be a rule of thumb.

5) You end the conclusion section with the desire for more collaboration: indeed, but this would also mean to stick to scientific argumentation and a proper criteria catalogue for comparison.

6) The structure is partly chaotic: Section 3.2 is squeezed between method and results, though I consider it as background knowledge.

Author Response

Thank you for the insightful suggestion from the reviewer. Please see the attachment.

Reviewer 2 Report

The paper entitled “Challenges of Applying Simplified LCA Tools in Sustainable 2 Design Pedagogy” aims at presenting the application of simple LCA tools in a university course.

The results show a positive potential of applying environmental innovation the tools to achieve function innovation in different products.

The authors should further discuss how environmental innovation can support sustainability through issues like the following ones:

  • Innovation strategies in a fruit growers association impacts assessment by using combined LCA and s-LCA methodologies
  • Innovative membrane filtration system for micropollutant removal from drinking water–prospective environmental LCA and its integration in business decisions
  • Does the potential of the use of LCA match the design team needs?
  • Environmental Innovation, Open Innovation Dynamics and Competitive Advantage of Medium and Large-Sized Firms

Furthermore, the authors should add a paragraph of paper’s limitations and caveats

Author Response

Thank you for the thoughtful suggestion from the reviewer. Please see the attachment.

Reviewer 3 Report

Dear Authors, 
I have found your work interesting and valuable.
In my opinion, it will allow the reader to learn new knowledge about eco-design methods and learn about design concepts and tools used for this purpose.
The work in terms of education has an interesting overtone.
My comments/suggestions for you concern the following issues: 
1. in work, you cite 29 references, but only 24% (7 out of 29) were published after 2011,
2. for example, in the sustainability-driven innovation area, it may be worth recalling the review article, e.g., Adams, R., Jeanrenaud, S., Bessant, J., Denyer, D., & Overy, P. (2016). Sustainability-oriented innovation: A systematic review. International Journal of Management Reviews, 18 (2), 180-205,          3. in the headings of tables 1 to 4, you provided the numbers of student groups in the following years (13 in 2015, 18 (2016), 10 (2017), and 7( 2018)). It would be worthy of providing how numerous these groups of students were.

Author Response

Thank you for the considerate suggestion from the reviewer. Please see the attachment.

Round 2

Reviewer 1 Report

Dear authors, thanks a lot for the revised version! You improved in many areas.

You improved on requested issues #1 reviewing literature, #2 innovation types and subtitles, #3-7 in most parts.

There are minor changes to consider:

  • Fig. 1: the image is blurred - could you improve that?
  • English: e.g. "Bright and Boks [9] stated that positive  designers’  opinion  on  LCA  use  in  Ecodesign  is  a  good  tool  in  several  parts  of Ecodesign, such as a selection of the problem areas and creative thinking." --> you may need minor language editing

Author Response

Thank you for the considerate suggestion from the reviewer. In this revision, we have accordingly addressed the issues raised by the reviewer. Please see the attachment. 
